# Gamma Secretase Inhibitors as Potential Therapeutic Targets for Notch Signaling in Uterine Leiomyosarcoma

**DOI:** 10.3390/ijms23115980

**Published:** 2022-05-26

**Authors:** Yasmin Abedin, Sofia Gabrilovich, Emily Alpert, Erica Rego, Salma Begum, Qingshi Zhao, Debra Heller, Mark H. Einstein, Nataki C. Douglas

**Affiliations:** 1Department of Obstetrics, Gynecology and Reproductive Health, Rutgers-New Jersey Medical School, Newark, NJ 07103, USA; s.d.gabrilovich@gmail.com (S.G.); emily.r.alpert@wustl.edu (E.A.); esr64@njms.rutgers.edu (E.R.); salmabeg2@gmail.com (S.B.); zhaoqi@njms.rutgers.edu (Q.Z.); hellerds@njms.rutgers.edu (D.H.); me399@njms.rutgers.edu (M.H.E.); nd537@njms.rutgers.edu (N.C.D.); 2Department of Pathology, Immunology and Laboratory Medicine, Rutgers-New Jersey Medical School, Newark, NJ 07103, USA

**Keywords:** notch signaling, uterine leiomyosarcoma, uterine cancer, γ-secretase inhibitor (GSI)

## Abstract

Uterine leiomyosarcoma (uLMS) is a rare and aggressive cancer with few effective therapeutics. The Notch signaling pathway is evolutionarily conserved with oncogenic properties, but it has not been well studied in uLMS. The purpose of our study was to determine expression of Notch family genes and proteins and to investigate the therapeutic effect of γ-secretase inhibitors (GSIs), indirect inhibitors of Notch signaling, in uLMS. We determined expression of Notch genes and proteins in benign uterine smooth muscle tissue, fibroids, and uLMS samples by immunostaining and in two uLMS cell lines, SK-UT-1B (uterine primary) and SK-LMS-1 (vulvar metastasis) by RT-PCR, Western blot and immunostaining. We exposed our cell lines to GSIs, DAPT and MK-0752, and measured expression of *HES1*, a downstream effector of Notch. Notch proteins were differentially expressed in uLMS. Expression of NOTCH3 and NOTCH4 was higher in uLMS samples than in benign uterine smooth muscle and fibroids. Expression of NOTCH4 was higher in SK-LMS-1 compared to SK-UT-1B. Exposure of SK-UT-1B and SK-LMS-1 to DAPT and MK-0752 decreased expression of *HES1* and decreased uLMS cell viability in a dose- and time-dependent manner that was unique to each GSI. Our findings suggest that GSIs are potential therapeutics for uLMS, albeit with limited efficacy.

## 1. Introduction

Uterine sarcomas are smooth muscle tumors that account for 3–7% of uterine cancers, with uterine leiomyosarcoma (uLMS) accounting for 65% of uterine sarcomas [1]. Although rare, uLMS is aggressive, with poor five-year survival rates of less than 50% in early-stage disease and survival rates of less than 15% in advanced stages [1,2,3,4]. Symptoms of uLMS, including abnormal uterine bleeding, an enlarged uterus, and/or pelvic pressure, mimic benign gynecologic conditions such as leiomyomata/fibroids of the uterus. Thus, diagnosis requires surgery and histologic evaluation [2,3].

Early-stage disease is treated with surgery, but recurrence rates are high, ranging between 50% and 70% [1,5,6]. Studies of treatment for early-stage disease with adjuvant chemotherapy have used ifosfamide, doxorubicin, gemcitabine, and docetaxel [1,5]. These studies have not shown a decrease in recurrence rates compared to no treatment; thus, the role of adjuvant therapy remains unclear [5]. Adjuvant radiotherapy has also not been shown to improve rates of disease recurrence or overall survival [1,7]. Adjuvant hormonal therapy has been considered as an option for uLMS that expresses hormone receptors, such as the estrogen receptor, which occurs in 25–80% of cases and/or the progesterone receptor, which is present in 30–75% of cases [1]. Thus, adjuvant therapies for early-stage disease have limited efficacy, and additional therapeutic approaches are needed.

Advanced stage disease is typically treated with chemotherapy, including doxorubicin, gemcitabine, and/or docetaxel, but efficacy is limited, toxicity is significant, and therapies are often poorly tolerated [1,6,8,9,10]. Translational research has focused on identifying and targeting dysregulated signaling pathways in various malignancies, including uLMS. Few therapeutic targets have been identified for uLMS, including inhibitors of receptor tyrosine kinase signaling and phosphatidylinositol 3-kinase/AKT/mammalian target of the rapamycin (PIK3/AKT/mTOR) pathway, with minimal benefits in clinical trials [11]. The Sonic Hedgehog pathway has also been shown to be upregulated and targetable in uLMS with some positive in vitro results [12]. Another pathway to be explored is the β-catenin/Wnt pathway, which has been shown to be upregulated and has many commercial inhibitors available [11]. The genetic make-up of uLMS has been investigated as well, with *TP53, RB1, ATRX*, and *PTEN* as the most common mutations, and can be used for the development of therapeutic targets [6]. Based on these insights, there is a critical clinical need for effective therapies for uLMS.

The Notch pathway is an evolutionarily conserved signaling pathway with oncogenic roles in many cancers, including colorectal cancer and non-small cell lung carcinoma [13]. The mammalian Notch pathway comprises four single-pass transmembrane receptors, NOTCH1, NOTCH2, NOTCH3, and NOTCH4, and five membrane-bound activating ligands, Delta-like ligand (DLL) 1, DLL3, and DLL4 and Jagged (JAG) 1 and JAG2. The binding of a ligand to the Notch receptor leads to the release of the Notch intracellular domain (NICD) via γ-secretase. The NICD translocates to the nucleus to regulate transcription of downstream effectors of Notch signaling, including members of the Hairy/Enhancer of Split (HES) and the Hairy/Enhancer of Split related with the YRPW motif (HEY) families and the Notch-regulated ankyrin repeat-containing protein (NRARP) (Figure 1) [14,15]. While the Notch signaling pathway has not been well studied in uLMS, it is implicated in other uterine cancers [14,15,16,17]. In endometrial adenocarcinoma, NOTCH1 was found to be highly expressed, while NOTCH4 had low expression [18]. It has been suggested that patients with endometrial cancer with higher NOTCH1 expression may have poorer prognoses [19].

γ-secretase inhibitors (GSIs) are indirect inhibitors of the Notch pathway (Figure 1). They inhibit the S3 ligand cleavage by γ-secretase, blocking signal transduction from a ligand-bound, activated receptor [20]. γ-secretase is an enzyme composed of four protein subunits, Presenilin (PS)1 or PS2, nicastrin, anterior pharynx defective-1 (APH-1)A or APH-1B, and presenilin enhancer 2 (PEN-2) [21]. DAPT (D-N-S-phenylglycine t-butyl ester) and MK-0752 (3-[4-(4-chlorophenyl)sulfonyl-4-(2,5-difluorophenyl) cyclohexyl] propanoic acid) are two GSIs currently being studied. DAPT binds to either PS1 or PS2 in the C terminal fragment of γ-secretase in the space between the docking site and the catalytic site of the enzyme. The drug is thought to prevent movement of the γ-secretase substrate (e.g., NOTCH receptor) through the enzyme [22]. MK-0752 is an aryl sulfone drug, the binding site of which remains unclear, but it likely involves binding to PS1 more selectively than to PS2. This may play a role in the less toxic profile of MK-0752 as compared to DAPT [23]. DAPT is well studied in vitro, but it is not used in vivo. MK-0752, an oral agent, has been shown to downregulate Notch signaling and inhibit cell growth in vitro in ovarian cancer models and to inhibit xenograft growth in vivo when used in combination with cisplatin [24]. MK-0752 has also been studied in phase I trials for pediatric soft tissue sarcomas and in combination with gemcitabine for pancreatic ductal adenocarcinoma. In these trials, MK-0752 was well tolerated with promising responses [20,25,26].

Due to the low rates of response to treatment, high rates of recurrence, and poor survival outcomes of uLMS, novel therapeutic agents are needed in the adjuvant, advanced-stage, and recurrent settings. Our objective was to determine if targeting Notch could be a novel therapeutic approach. We hypothesized that DAPT and MK-0752 would decrease Notch signaling in uLMS and reduce key features of tumor progression. We first determined endogenous Notch expression in uLMS patient tissue samples and uLMS cell lines as compared to benign tissue samples and cell lines, respectively. We then treated uLMS cell lines with DAPT and MK-0752 using varying time points and concentrations of each GSI. We determined how exposure to GSIs affected uLMS cell viability and Notch signaling activity. Lastly, we treated uLMS cell lines with DAPT and MK-0752 to determine the effects of γ-secretase inhibition on uLMS cell proliferation and invasion. The data presented herein demonstrate a potential role for GSIs and Notch inhibition in the treatment of uLMS.

## 2. Results

### 2.1. Differential Expression of Notch Proteins in uLMS

To determine expression of Notch in benign uterine tissues and uLMS, we collected uterine tissue samples at the time of hysterectomy for benign uterine pathology (BH) or for uterine fibroids (UF) and purchased a LMS tissue array that included 30 uLMS tissue samples (TA) (Appendix A). Internal pathology review of the TA confirmed uLMS and revealed that there was a range of mitotic factors, atypia, and tumor necrosis among the samples (Appendix A) [2]. H&E staining revealed that BH tissue contained glands, vasculature, and smooth muscle, while UF tissue and uLMS TA samples contained smooth muscle and vasculature (Figure 2A–C). Immunostaining was performed to determine expression of NOTCH1, NOTCH3, and NOTCH4 and to quantify the amount of vasculature, as detected by expression of CD31, in each sample. We were unable to determine expression of NOTCH2, as all antibodies we used gave non-specific staining in our control tissues. We found that NOTCH1 and NOTCH4 were predominantly expressed in the vasculature in all tissue samples (Figure 2H–J,P–R). Expression of CD31, NOTCH1, and NOTCH4 was measured as the amount of positive staining per tissue area. Expression of CD31 was significantly higher in uLMS TA samples compared to BH samples but was similar to UF samples (Figure 2G), suggesting increased vasculature in uLMS samples as compared to benign uterine smooth muscle. Whereas expression of NOTCH1 was similar in BH, UF, and TA samples (Figure 2H–K), expression of NOTCH4 was significantly higher in TA samples compared to UF samples but was similar to BH samples (Figure 2S). For NOTCH3, H-score was used to quantify expression. NOTCH3 was expressed in smooth muscle cells, and we found that expression of NOTCH3 was significantly higher in TA samples compared to BH samples, but similar to expression in UF samples (Figure 2L–O). These results suggest that uLMS samples have more vessels and increased expression of NOTCH3 and NOTCH4 compared to benign uterine tissues.

### 2.2. uLMS Cell Lines Are Distinct in Morphology and Notch Expression

We characterized expression of Notch family mRNAs and proteins in HUt-SMC, SK-UT-1B, and SK-LMS-1. HUt-SMC is a fibroblast smooth muscle cell line. SK-UT-1B is a primary uLMS cell line with epithelial-like morphology and is smaller in size compared to HUt-SMC and SK-LMS-1 (Figure 3A,B,D,E). SK-LMS-1 is a vulvar metastasis of uLMS and has a fibroblast morphology (Figure 3C,F).

Using RT-PCR, we found that HUt-SMC and SK-UT-1B expressed *NOTCH1*, *NOTCH2*, *NOTCH3*, and *NOTCH4.* SK-LMS-1 cells expressed *NOTCH1*, *NOTCH2*, and *NOTCH4*, and had minimal to no expression of *NOTCH3* (Figure 3G). Expression of Notch proteins was determined by Western blot (Figure 3H). Consistent with the mRNA expression, we detected expression of all four Notch proteins in HUt-SMC cells. Detection of N1ICD, N3ICD, and N4ICD in HUt-SMC indicated active Notch signaling via the NOTCH1, NOTCH3, and NOTCH4 receptors in HUt-SMC [27,28,29]. While detection of the NOTCH2 transmembrane portion (N2TM) indicated NOTCH2 protein expression, it did not indicate NOTCH2 signaling activity in HUt-SMC [30]. SK-UT-1B cells expressed N1ICD, which indicated active Notch signaling via NOTCH1, but did not express N4ICD/NOTCH4. For SK-LMS-1, detection of N1ICD and the intense band for N4ICD confirmed expression of NOTCH1 and NOTCH4 proteins with signaling via these receptors. Detection of N2TM in both SK-UT-1B and SK-LMS-1 indicated expression of the NOTCH2 protein without confirmation of NOTCH2 signaling activity. Detection of a non-specific signal in SK-UT-1B and SK-LMS-1 for NOTCH3 indicated that NOTCH3 was not expressed in these cells. Using IF staining, we detected expression of HES1, an effector of Notch signaling (Figure 1), in HUt-SMC, SK-UT-1B, and SK-LMS-1 cells (Figure 3I–K). Together, these data suggest that active Notch signaling is mediated by different Notch proteins in HUt-SMC, SK-UT-1B, and SK-LMS-1 cells.

### 2.3. DAPT and MK-0752 Decrease uLMS Cell Viability

MTT assays were performed to determine the half-maximal inhibitory concentration (IC_50_) after 24 h of exposure to DAPT and MK-0752. Exposure of SK-UT-1B cells to GSIs yielded an IC_50_ of 90.13 µM for DAPT and 128.4 µM for MK-0752 (Figure 4A,B). Exposure of SK-LMS-1 cells to GSIs yielded an IC_50_ of 129.9 µM for DAPT and 427.4 µM for MK-0752 (Figure 4C,D). We then calculated the IC_5_ and IC_30_ concentrations for each condition. For SK-UT-1B cells, the IC_5_ and IC_30_ concentrations were 15 and 50 µM for DAPT and 50 and 95 µM for MK-0752, respectively. For SK-LMS-1 cells, the IC_5_ and IC_30_ concentrations were 20 and 65 µM for DAPT and 50 and 100 µM for MK-0752, respectively. These data show that increasing concentrations of both DAPT and MK-0752 increase the death of uLMS cells.

### 2.4. GSIs Inhibit Notch Signaling in uLMS Cells in a Time-Dependent and Dose-Dependent Manner

To determine if inhibition of γ-secretase decreased Notch signaling in uLMS cells, we exposed cells to inhibitory concentrations of DAPT and MK-0752 at which 95% (IC_5_) and 70% (IC_30_) of the cells were viable. After each cell line was exposed to the GSI for either 4 or 24 h, we performed qPCR to determine mRNA expression of *HES1.* There was no change in expression of *HES1* in SK-UT-1B after 4 h with DAPT at IC_5_ of 15 µM or IC_30_ of 50 µM (Figure 5A,C) or with MK-0752 at IC_5_ of 50 µM or IC_30_ of 95 µM (Figure 5B,D). Expression of *HES1* in SK-UT-1B only decreased after 24 h exposure to DAPT at the higher concentration, IC_30_ of 50 µM (FC: 1.55, *p* ≤ 0.01) (Figure 5G). After 24 h, there was no change in expression of *HES1* with DAPT at IC_5_ of 15 µM (Figure 5E) or with MK-0752 at the IC_5_ of 50 µM or IC_30_ of 95 µM (Figure 5F,H). Expression of *HES1* decreased in SK-LMS-1 after 4 h with DAPT at IC_5_ of 20 µM (FC:1.46, *p* ≤ 0.01) and IC_30_ of 60 µM (FC:1.74, *p* ≤ 0.01) (Figure 5I,K), but not with MK-0752 at the IC_5_ of 50 µM or IC_30_ of 100 µM (Figure 5J,L). Expression of *HES1* in SK-LMS-1 decreased after 24 h with DAPT at IC_5_ of 20 µM (FC: 3.17, *p* ≤ 0.01) and IC_30_ of 60 µM (FC: 1.40, *p* ≤ 0.05) (Figure 5M,O) and with MK-0752 at IC_5_ of 50 µM (FC: 1.60, *p* ≤ 0.05) and IC_30_ of 100 µM (FC:4.17, *p* ≤ 0.01) (Figure 5N,P). These data showed that a 4 h exposure of SK-UT-1B to either DAPT or MK-0752 did not alter Notch signaling. In contrast, 4 h exposure of SK-LMS-1 to DAPT, but not MK-0752, significantly decreased Notch signaling. With a 24 h exposure, Notch signaling in SK-LMS-1 was significantly decreased by both DAPT and MK-0752. Exposure to the highest concentration of DAPT for 24 h was the only condition in which a GSI decreased Notch signaling in SK-UT-1B. Together, these findings show that GSI exposure inhibits Notch signaling in a time-dependent and dose-dependent manner, which is important to consider when utilizing these agents for further studies.

### 2.5. Exposure to GSIs Does Not Impact uLMS Cellular Proliferation or Invasion

Properties of cancer cells contributing to metastasis are the cell’s ability to proliferate and invade surrounding tissues. To determine if DAPT or MK-0752 decreased cellular proliferation and/or invasion in uLMS cell lines, we assessed proliferation and invasion after 24 h exposure to the lowest dose at which a GSI inhibited Notch signaling. For SK-UT-1B, proliferation and invasion assays were performed at the IC_30_ with exposure to 50 µM DAPT. No difference in proliferation or invasion was observed (Appendix A). Given that exposure of SK-UT-1B to MK-0752 did not decrease HES1, we did not assess proliferation or invasion. For SK-LMS-1, proliferation and invasion assays were performed at the IC_5_ with exposure to 20 µM DAPT or 50 µM MK-0752. Exposure to DAPT or MK-0752 at the IC_5_ did not impact cellular proliferation or invasion (Appendix A). Taken together, our findings demonstrate that DAPT is more effective than MK-0752 at reducing Notch signaling activity in uLMS cells, but neither GSI decreases uLMS cell proliferation or invasion, which are required for tumor progression.

## 3. Discussion

To determine if γ-secretase inhibitors can be used as potential therapeutic targets for Notch signaling in uLMS, we investigated the expression and inhibition of Notch signaling. We demonstrated expression of Notch proteins in benign uterine tissue, uLMS tissue samples, and uLMS cell lines SK-UT-1B and SK-LMS-1. We showed that Notch signaling is active in uLMS cell lines and that inhibition of γ-secretase with both DAPT and MK-0752 inhibits Notch signaling and decreases uLMS cell viability. To the best of our knowledge, this is the first study that determined Notch expression in uLMS tissue samples and showed that Notch signaling is active in uLMS, making the Notch signaling pathway a potential therapeutic target.

Cell-type specific expression of Notch proteins in the human uterine tissues has not been well described. In this study, we collected uterine tissue at the time of gynecologic surgery for benign pathology. All BH samples were from patients with uterine fibroids. Two of the patients also had adenomyosis, or abnormal glandular tissue within the myometrium (Appendix A), which we excluded in our analyses of the myometrium [31,32]. Based on mouse studies, Notch1 and Notch4 proteins are expressed in the vasculature in the endometrium. Notch1 is also expressed in the myometrial vasculature, while Notch2, Notch3, and Notch4 are expressed in uterine smooth muscle cells [33,34]. Notch expression in our human uterine and uLMS tissues was consistent with expression in the mouse uterus. With quantification, we found higher NOTCH3 expression in the smooth muscle of TA samples compared to BH samples and higher NOTCH4 expression in TA samples compared to UF samples, suggesting that expression levels of NOTCH3 and NOTCH4 could be used to measure disease progression in uLMS [33]. CD31, a known vascular endothelial cell marker, was highly expressed in TA samples, suggesting that uLMS is a highly vascular malignancy capable of angiogenesis, which could be one explanation for its aggressive nature [34]. As NOTCH1 and NOTCH4 were expressed in uterine vasculature, we expected higher NOTCH1 and NOTCH4 expression in uLMS TA samples compared to BH samples, paralleling the higher CD31 expression in uLMS TA samples compared to BH samples. However, that was not what we observed. Levels of NOTCH1 and NOTCH4 were similar in uLMS and BH samples. Perhaps the combination of NOTCH1 and NOTCH4 expression, as opposed to expression of either protein alone, would be significantly higher in uLMS TA samples compared to BH tissue. We could not test this with co-staining because the antibodies used to detect expression of NOTCH1 and NOTCH4 were raised in the same host species.

The cell lines SK-UT-1B and SK-LMS-1 were both derived from uLMS tumors, however, SK-UT-1B is a primary uLMS cell line with epithelial morphology, while SK-LMS-1 is from a vulvar metastasis of uLMS and has a fibroblast morphology. The differences in Notch expression we observed may reflect the anatomical site and/or the stage of disease from which the cell line was derived. HUt-SMC expressed NOTCH1 and NOTCH3 most intensely, which is consistent with studies showing Notch3 in uterine smooth muscle cells [33]. SK-UT-1B expressed NOTCH1 most intensely, while SK-LMS-1 expressed NOTCH4 most intensely. Given the expression of NOTCH3 in the uLMS TA samples, we were surprised to find that our uLMS cells did not express this Notch protein. This difference could have been observed for a few reasons. uLMS patient tissue samples are heterogenous, while uLMS cell lines represent one sample. In addition, in vivo expression of NOTCH3 expression may be upregulated by signals from other cell types. To the best of our knowledge, NOTCH4 expression has not been previously described in the human myometrium. We detected high expression of NOTCH4 in SK-LMS-1 cells and in uLMS TA samples. Therefore, targeting NOTCH4 with monoclonal antibodies could be explored as a potential therapeutic option for uLMS.

Expression of the cleaved intracellular domains of NOTCH1 and NOTCH4 and nuclear expression of HES1 in both SK-UT-1B and SK-LMS-1 demonstrated active Notch signaling. Our lab has previously shown in cell lines expressing multiple Notch proteins that decreased mRNA expression of *HES1* after exposure to DAPT reflects decreased Notch signaling [35]. In uLMS cells, DAPT and MK-0752 decreased Notch signaling in a dose- and time-dependent manner. *HES1* was decreased in SK-UT-1B with the higher concentration (IC_30_) of DAPT at 24 h, but not at the lower concentration (IC_5_) or at 4 h. There was no decrease in Notch signaling in SK-UT-1B when treated with MK-0752. SK-LMS-1 was sensitive to DAPT at both 4 h and 24 h and at both IC_5_ and IC_30_. However, Notch signaling was only decreased in SK-LMS-1 treated with MK-0752 at 24 h. The concentrations we used are scalable to 3D cell culture models, mouse models, and human trials [24,26,36]. Moreover, the time dependence of Notch inhibition gives insight into the metabolism of MK-0752, which can help determine dose schedules for in vivo studies.

We showed that uLMS cells respond differently to treatment with two GSIs. DAPT decreased Notch signaling more often than MK-0752 and did so in both cell lines. This not only suggested greater efficacy of DAPT, but also highlighted the lesser specificity of DAPT as compared to MK-0752. DAPT binds without preference to both PS1 and PS2 (Figure 1), which contributes to intolerability in humans because of gastrointestinal side effects. MK-0752 binds more specifically to PS1, making it more tolerable, but less effective [22,23]. We had hypothesized that inhibition of γ-secretase would decrease uLMS proliferation and/or invasion, but this was not observed. Neither proliferation nor invasion was affected by exposure to concentrations of DAPT and MK-0752 that decreased Notch signaling. Given the limited efficacy of GSIs, combinations of GSIs with common cytotoxic agents for uLMS may have greater efficacy in decreasing tumor cell viability and functions such as proliferation and invasion that are required for tumor progression. Prior studies examined the effects of GSIs in combination with gemcitabine to treat pancreatic ductal adenocarcinoma (PDAC) [26,37]. In a phase I trial, MK-0752 was combined with gemcitabine for PDAC and was found to be well tolerated with some gastrointestinal side effects. Most patients receiving combination therapy achieved stable disease, with one patient achieving partial response. Another study investigated the effects of a GSI (LY3039478) alone in soft tissue sarcomas, including leiomyosarcoma but not specifically of uterine origin, and found modest response and tolerable side effects [38]. Future studies will determine how the combination of GSIs, such as MK-0752, with gemcitabine or other common chemotherapeutics used to treat uLMS impact cell viability [8,9,10].

There is a clinical need for new treatment options for uLMS. Our study is unique in that we identified a potential targetable pathway for this rare gynecologic malignancy. Hemming et al. explored gene expression by RNA sequencing of LMS compared to various sarcomas. Few uterine tumors were included, limiting conclusions about uLMS and targetable signaling pathways [39]. We found that NOTCH4 is upregulated in SK-LMS-1 and in uLMS tissue compared to UF tissue, which means that NOTCH4 can possibly be tested as a new diagnostic and/or prognostic marker for uLMS. Our study also identified GSIs as potential therapeutics against uLMS. Fostering this, GSIs could be used in combination with common chemotoxic agents for uLMS to facilitate maximal tumor response. Moreover, other targets of γ-secretase and targets upstream and downstream of γ-secretase can potentially be studied to treat uLMS. Thus, our work advances knowledge about LMS and explores potential targetable therapies.

Our study has additional strengths and some limitations. A strength of this work is the evaluation of two different uLMS cell lines, SK-UT-1B (primary uLMS with epithelial cellular morphology) and SK-LMS-1 (vulvar metastasis of a uLMS tumor with fibroblast cellular morphology). We showed that each uLMS tumor and subtype respond distinctively to treatment. We further examined the impact of two GSIs of different molecular classes, one of which has been studied in clinical trials as an oral medication, potentiating the use of this drug. Another strength of our study is that we characterized the expression pattern of Notch proteins in human myometrium, which lays the foundation for investigating how dysregulation of the Notch pathway is related to uLMS initiation. While our study is unique in that we were able to examine many uLMS human tissue samples from a leiomyosarcoma tissue array in which 30 of 80 samples were uLMS, we did not have available fresh uLMS tissue to confirm with immunostaining and to use for murine models. Because diagnosis of uLMS prior to surgery and histologic evaluation is infrequent due to it often being an unexpected diagnosis, studies involving fresh uLMS tissue or patient-derived xenograft (PDX) murine models are rare. Other limitations of our study include the small number of benign uterine tissue samples and small size (1.5 mm diameter) of each uLMS TA sample, especially when compared to the benign uterine samples (4–8 mm). We acknowledge that a uLMS TA sample likely does not represent the full heterogeneity of the patient’s tumor, limiting the generalizability of our results. Each BH sample was collected at a different phase of the menstrual cycle and with different pathologies, which could alter the endogenous Notch protein expression. Given the limitation of small sample sizes of our uterine benign tissues, all data were compared using non-parametric statistical tests so as not to augment differences we observed. Lastly, we encountered technical limitations, which prevented localization of NOTCH2 expression via immunostaining, thus limiting our conclusions about the comprehensive expression and activity of Notch signaling in benign uterine tissues and patient uLMS samples.

Despite these limitations, our study has addressed some knowledge gaps regarding the relationship of the Notch pathway and uLMS that will facilitate further investigation of novel therapies for uLMS. Herein, we used immunostaining to show the localized expression of NOTCH1, NOTCH3, and NOTCH4 in uLMS human samples as compared to benign myometrial and fibroid tissue. We demonstrated endogenous expression of Notch proteins in uLMS cell lines via Western Blot, with NOTCH1 being intensely expressed in SK-UT-1B and NOTCH4 being most highly expressed in SK-LMS-1. Further, we identified differences in efficacy between DAPT and MK-0752, with DAPT being more efficacious. We also determined the time dependence of GSIs and the need to expose uLMS cells to GSIs for longer durations in order to inhibit the Notch pathway.

Some gaps still exist. Progress toward novel therapies requires in vivo experiments to first determine the efficacy of GSIs in animal models. Although it is difficult to obtain primary uLMS tissue, SK-LMS-1 has previously been used in mouse xenograft models and can be used for initial in vivo studies in the future [40]. These in vivo studies will provide additional data that can lead to clinical trials using GSIs in combination with chemotherapeutics for uLMS. Future studies can inhibit expression of specific Notch receptors and/or ligands in uLMS, utilizing monoclonal antibodies or CreLoxP recombination in murine models. Addressing some of the remaining knowledge gaps will aid in further understanding the function of the Notch pathway in uLMS and how Notch can be targeted as an adjuvant therapy.

In conclusion, we showed that Notch signaling was active, and Notch genes and proteins were differentially expressed in uLMS. GSIs decreased Notch signaling in uLMS in a dose- and time-dependent manner. Taken together, these findings have identified the Notch signaling pathway as a potential therapeutic target for uLMS. MK-0752, which is orally tolerated, should be investigated in 3D in vitro and in vivo models, either alone or in combination with other cytotoxic agents, to further assess efficacy as a potential therapeutic modality against uLMS. The combination of MK-0752 with doxorubicin, gemcitabine, or docetaxel, which are common chemotherapeutics for uLMS, is a novel treatment strategy that may be required for maximal tumor response to therapy.

## 4. Materials and Methods

### 4.1. Patient Samples

The study protocol (Pro2019000723) was approved by the institutional review board at Rutgers New Jersey Medical School for collection of patient samples after obtaining written signed informed consent. Benign hysterectomy (BH) patient tissue (*n* = 3) and uterine fibroid (UF) patient tissue (*n* = 3) were collected at the time of surgery and were used as controls. Paraffin-embedded tissue was cut into 7 µm sections for BH and UF samples. Sample characteristics are in Appendix A. uLMS samples (*n* = 30) were from a leiomyosarcoma tissue array (TA) (US Biomax, Inc., Derwood, MD, USA, Serial No. SO8013). As per the manufacturer, paraffin-embedded uLMS samples were cut at 5 µm and 1.5 mm in diameter. Each sample from the TA was reviewed by the Rutgers New Jersey Medical School pathologist. The schematic layout of the tissue array and characterization of the uLMS samples are presented in Appendix A.

### 4.2. Cell Lines and Cell Culture

Two human uLMS cancer cell lines, SK-UT-1B (ATCC HTB-115™) and SK-LMS-1 (ATCC HTB-88™), and one human uterine smooth muscle cell line HUt-SMC (ATCC PCS-460-011™), were purchased from ATCC (Manassas, VA, USA). uLMS cell lines were cultured in Eagle’s Minimum Essential Medium (EMEM) (ATCC) with 10% Fetal Bovine Serum (FBS) and 1% Pen-Strep. HUt-SMC was cultured in Vascular Smooth Muscle Cell Growth Kit (ATCC) as per the manufacturer’s instructions. All cells were maintained in a humidified incubator under standard culture conditions of 21% O_2_ and 5% CO_2_ at 37 °C.

γ-secretase activity was blocked by DAPT (Sigma-Aldrich, St. Louis, MO, USA) or MK-0752 (MedChemExpress, Monmouth, NJ, USA) dissolved in DMSO. SK-UT-1B and SK-LMS-1 cells were grown to greater than 80% confluence in 60 mm plates or 6-well plates and treated with DAPT or MK-0752. Vehicle control (DMSO) was used at equal volume to treatment.

### 4.3. Immunostaining

Paraffin-embedded BH, UF, and TA were sectioned and stained with hematoxylin and eosin (H&E) using standard methods. Immunostaining was performed to detect expression of CD31, NOTCH1, NOTCH3, NOTCH4, HES1, and Ki-67. Sections stained with IgG to the same concentration as the primary antibody served as negative controls. Tissue sections were deparaffinized, and slides were baked in 0.01M citrate buffer (Poly Scientific R&D Corp, Bay Shore, NY, USA) for antigen retrieval (2100 Retriever, BioVendor, Brno, Czech Republic). For immunofluorescence (IF), tissue was incubated in primary antibody overnight at 4 °C and in secondary antibody for 1 h at room temperature. For IF on Hut-SMC, SK-UT-1B, and SK-LMS-1, cells were grown on coverslips, fixed with 4% paraformaldehyde (PFA), and then stained as described above without antigen retrieval. Cells stained with the secondary antibody alone served as negative controls. Vectashield containing 40, 6-diamidino-2-phenylindole (DAPI, Vector, Burlingame, CA, USA) was used for nuclear visualization and mounting.

For immunohistochemistry (IHC), BH, UF, and TA tissue samples were incubated with primary antibody overnight at 4 °C followed by the secondary antibody for 30 min at room temperature. Sections stained with IgG served as negative controls. Slides were developed with the Vectastain ABC kit (Vector), DAB substrate kit (Vector), and mounted using Permount (Fisher Scientific, Fairlawn, NJ, USA). Primary and secondary antibodies used for immunostaining are listed in Appendix A.

Images were taken on the Keyence BZ-X710 All-in-One Fluorescent Microscope (Keyence) (Keyence, Osaka, Japan). All images were analyzed on ImageJ Fiji [41]. To determine expression of N1ICD, CD31, and NOTCH4, percent positive staining over section area was quantified. For all BH samples, myometrial tissue was scored, and glands and stroma were excluded. To quantify NOTCH3 expression, the H-score was determined using standard methodology [42]. The percentage of high-intensity staining area was multiplied by 3, percentage of mid-intensity staining area multiplied by 2, percentage of low-intensity staining area multiplied by 1, and percentage of no staining area multiplied by 0. These scores were then added for a total score ranging between 0 and 300.

### 4.4. RNA Extraction, Real Time-PCR (PCR), and Quantitative RT-PCR (qPCR)

Total RNA was extracted using Rneasy Mini Kit (Qiagen, Germantown, MD, USA) for Hut-SMC, SK-UT-1B, and SK-LMS-1 and reverse transcribed into cDNA using qScript cDNA Supermix (Quanta Bio, Beverly, MA, USA). After 35 cycles of amplification, PCR products underwent agarose gel visualization with *18s* rRNA as loading control. Relative gene expression was determined by qPCR with QuantiNova SYBR Green PCR Kit (Qiagen) in triplicate for a total of 6 samples. Relative expression level of each target gene was quantified by fold change (2^−ΔΔCt^) as compared to *18s* rRNA expression. Primers are listed in Appendix A.

### 4.5. Western Blot

Whole cell lysates of Hut-SMC, SK-UT-1B, and SK-LMS-1 were prepared with RIPA lysis buffer, and concentrations were determined by Pierce BCA protein assay kit (Thermo Fisher Scientific, Waltham, MA, USA) with bovine serum albumin (BSA) as standard. For each cell line, 20 µg of protein was denatured and fractionated by Sodium dodecyl-sulfate polyacrylamide gel electrophoresis (SDS-PAGE), transferred to nitrocellulose membrane, probed with primary antibody followed by HRP conjugated secondary antibody (Invitrogen, Waltham, MA, USA), and expression detected by enhanced chemiluminescence. Expression of α-tubulin served as loading control. Primary and secondary antibodies are listed in Appendix A.

### 4.6. Inhibitory Concentrations of DAPT and MK-0752

An MTT (3-(4,5-dimethylthiazol- 2-yl)-2,5-diphenyltetrazolium bromide) assay kit (Abcam, Cambridge, UK) was used to determine cell viability after treatment of SK-UT-1B and SK-LMS-1 with DAPT or MK-0752 at various concentrations. Cells were plated in a 96-well plate and allowed to attach. Cells were then treated with media containing 10–500 µM DAPT or 10–500 µM MK-0752 or DMSO (vehicle control) for 24 h. Cells were then incubated with 50 mL of freshly diluted 0.5 mg/mL MTT solution for 4 h at 37 °C. Thereafter, 150 mL of DMSO was added to each well, followed by incubation for another 30 min at 37 °C, and absorbance was measured at 570 nm in a microtiter plate reader. Dose–response curves were generated, and inhibitory concentrations (IC) for IC_5_, IC_30_, and IC_50_ were determined using GraphPad Prism Version 9.0 (San Diego, CA, USA).

### 4.7. Proliferation Assay

SK-UT-1B and SK-LMS-1 cells treated with DAPT, MK-0752, or DMSO for 24 h were fixed with 4% PFA, stained for Ki-67 (Abcam) expression, and counterstained with DAPI (Vector). Cells were imaged on the Keyence. Images were analyzed using ImageJ Fiji [41]. Two independent observers counted the number of Ki-67 positive cells and the total number of DAPI positive cells in 10× magnified slides, for at least *n* = 6. Percentage of proliferating cells was determined as the number of Ki-67 positive cells per total cell count × 100. Proliferation after DAPT or MK-0752 treatment was normalized to the DMSO vehicle control.

### 4.8. Transwell Invasion Assay

Invasion assays were performed in 24-well transwell chambers (Corning, Corning, NY, USA). Falcon transparent cell culture inserts (Corning, 8 µm pore size) were coated with growth-factor reduced, phenol red-free Matrigel^TM^ (Corning) mixed with media at a final concentration of 0.3 mg/mL (thick coating) and 0.03 mg/mL (thin coating) and dried for two h at 37 °C. SK-UT-1B and SK-LMS-1 cells were serum-starved overnight before resuspension in serum-reduced media (0.1% FBS) with DAPT, MK-0752, or DMSO and added to the upper compartment. Chemoattractant media (10% FBS) was added to the lower compartment. After 24 h, cells on the undersurface of the membrane were fixed with 4% PFA and rinsed with PBS. The membrane was excised with a scalpel, mounted on a glass slide with DAPI (Vector Laboratories), and a coverslip placed on top. Slides were imaged using the Keyence at 10× magnification. Images were analyzed with ImageJ Fiji [41], and two independent observers counted the total number of cells on the undersurface of each membrane. Percent invasion was represented as number of cells invading through the thick Matrigel layer (0.3 mg/mL) divided by the number of cells invading through the thin Matrigel layer (0.03 mg/mL). Data were normalized to the DMSO vehicle control.

### 4.9. Statistical Analysis

Medians were compared using Mann–Whitney U test or Kruskal–Wallis test with Dunn post hoc testing. Data are presented as median with interquartile range (median + IQR) and reported as fold change (FC) with *p* values. Statistical analyses were performed using GraphPad Prism Version 9.0. Statistical significance was defined as *p* ≤ 0.05.

## Figures and Tables

**Figure 1 ijms-23-05980-f001:**
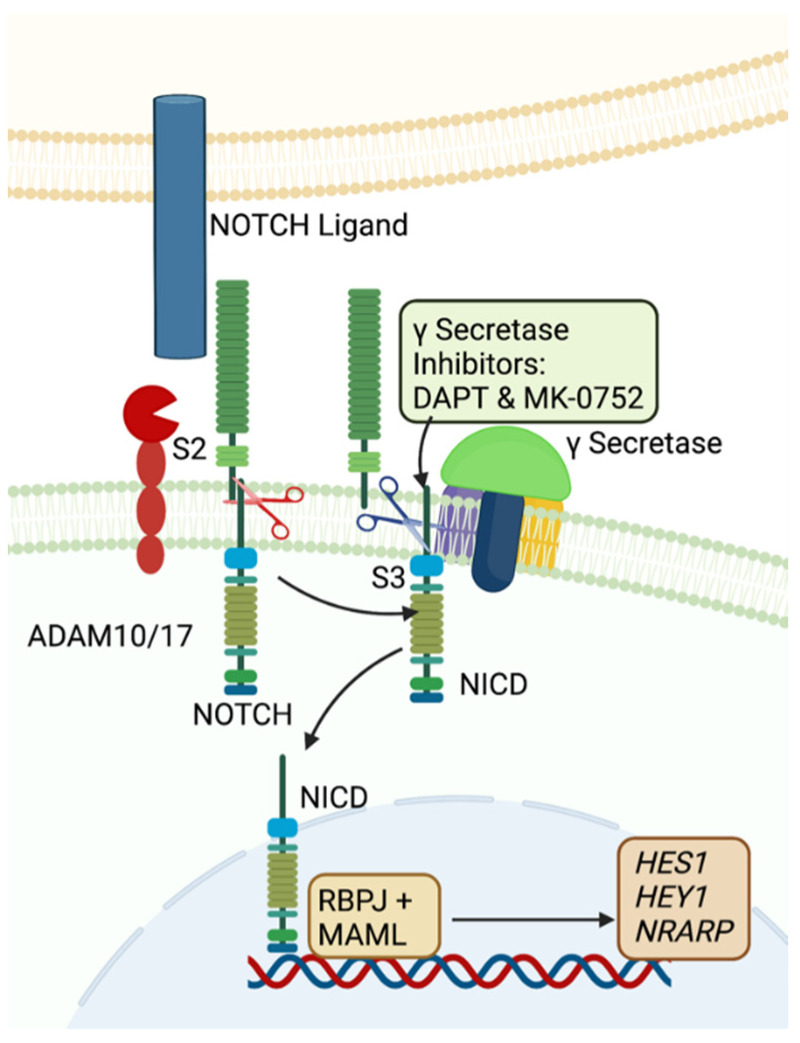
Schematic representation of the Notch signaling pathway. Notch signaling is activated when a NOTCH ligand binds to a NOTCH receptor on an adjacent cell. This triggers ADAM10/17 to perform an S2 cleavage of the NOTCH receptor. γ-secretase then catalyzes an S3 cleavage of the receptor, which releases the Notch intracellular domain (NICD) from the cell membrane. The NICD translocates to the nucleus and forms a complex with DNA-binding transcription factors such as Recombination Signal Binding Protein for Immunoglobulin Kappa J Region (RBPJ) and Mastermind Like (MAML), resulting in transcription of downstream effectors such as *HES1*. When cells are exposed to γ-secretase inhibitors (GSIs), such as DAPT or MK-0752, S3 cleavage of the NOTCH receptor does not occur, preventing release of the NICD. Created with BioRender.com.

**Figure 2 ijms-23-05980-f002:**
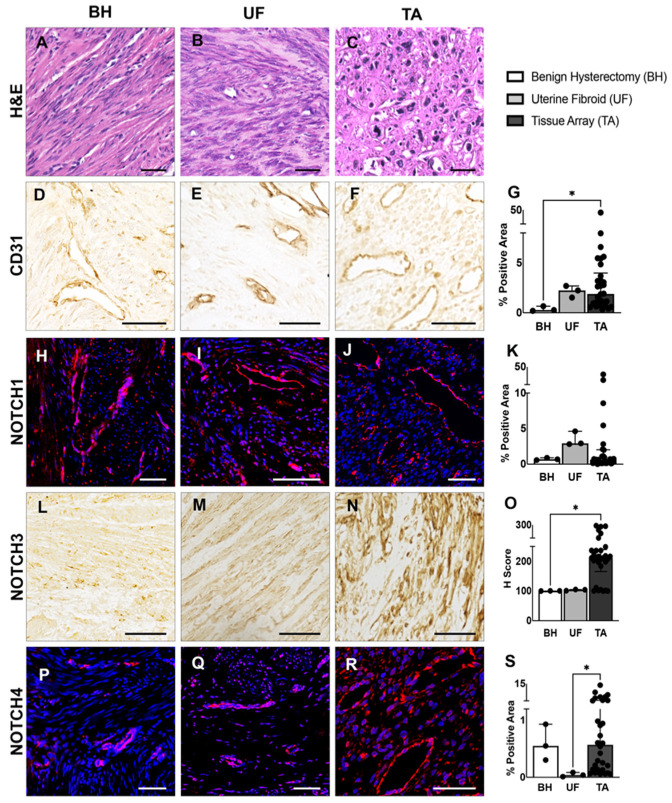
Notch expression in benign uterine tissues and uLMS. (**A**–**C**) Representative H&E images of uterine tissue from hysterectomies for benign pathology (BH, *n* = 3), uterine fibroids (UF, *n* = 3), and uLMS tissue array samples (TA, *n* = 30). (**D**–**F**) Representative images of CD31 immunostaining. (**G**) Expression of CD31 in uLMS TA samples was higher than in BH samples (* *p* < 0.05). (**H**–**J**) Representative images of immunostaining for NOTCH1. (**K**) Expression of NOTCH1 was similar in BH, UF, and TA tissue samples. (**L**–**N**) Representative images of immunostaining for NOTCH3. (**O**) Expression of NOTCH3 was higher in uLMS TA samples than in BH samples (* *p* < 0.05). (**P**–**R**) Representative images of immunostaining for NOTCH4. (**S**) Expression of NOTCH4 in uLMS TA samples was significantly higher than in UF samples (* *p* < 0.05). Data are represented as median + IQR. Scale bars = 50 µm.

**Figure 3 ijms-23-05980-f003:**
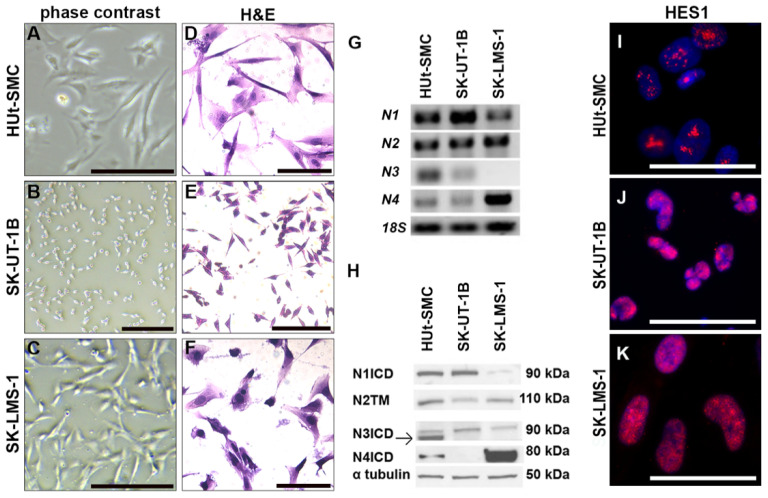
Notch expression in cell lines. (**A**–**C**) Representative phase contrast images of each cell line. (**D**–**F**) Representative H&E stained images of each cell line. (**G**) mRNA expression of *NOTCH* genes *(N* = *NOTCH)* by RT-PCR. *18s* rRNA was loading control. HUt-SMC and SK-UT-1B expressed *N1*, *N2*, *N3*, and *N4.* SK-LMS-1 cells expressed *N1*, *N2*, and *N4*, and minimal *N3.* (**H**) Expression of NOTCH proteins *(N = NOTCH)* by Western Blot. ⍺-tubulin was loading control. N1ICD, N3ICD and the N4ICD were expressed in HUt-SMC, indicative of active Notch signaling. The transmembrane region of NOTCH2 (N2TM) was expressed in HUt-SMC. Expression of N2TM demonstrated expression of NOTCH2 protein, but not NOTCH2 signaling activity in HUt-SMC. SK-UT-1B cells expressed N1ICD. SK-LMS-1 expressed N4ICD. The antibody used to detect N3ICD yielded non-specific signals in SK-UT-1B and SK-LMS-1. (**I**–**K**) Representative images of immunostaining for HES1. Scale bars = 200 µm in (**A**,**C**); 100 µm in (**B**,**D**–**F**); and 50 µm in (**I**–**K**).

**Figure 4 ijms-23-05980-f004:**
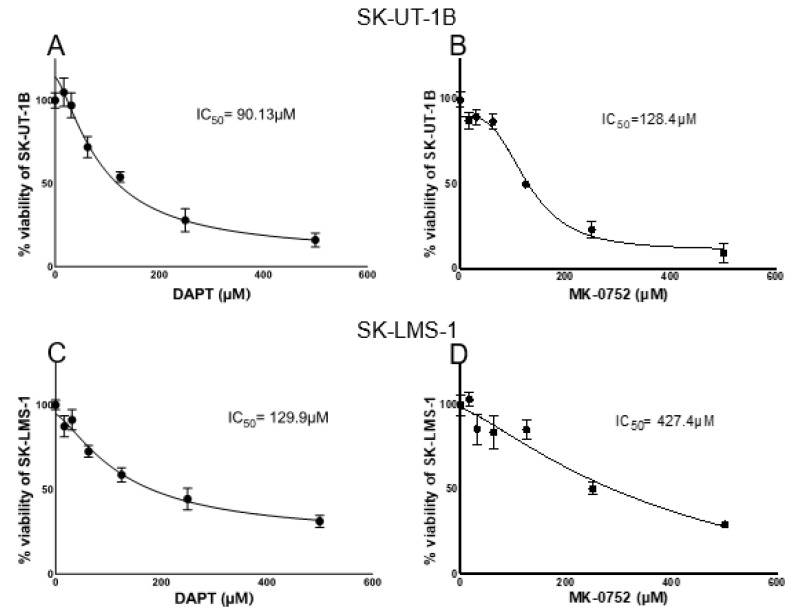
Dose–response curves for inhibitory concentrations. Half maximal inhibitory concentrations (IC_50_) were determined and utilized to determine IC_5_ and IC_30_ used for treatment. Graphs of SK-UT-1B treated with DAPT (**A**) and MK-0752 (**B**) and SK-LMS-1 treated with DAPT (**C**) and MK-0752 (**D**).

**Figure 5 ijms-23-05980-f005:**
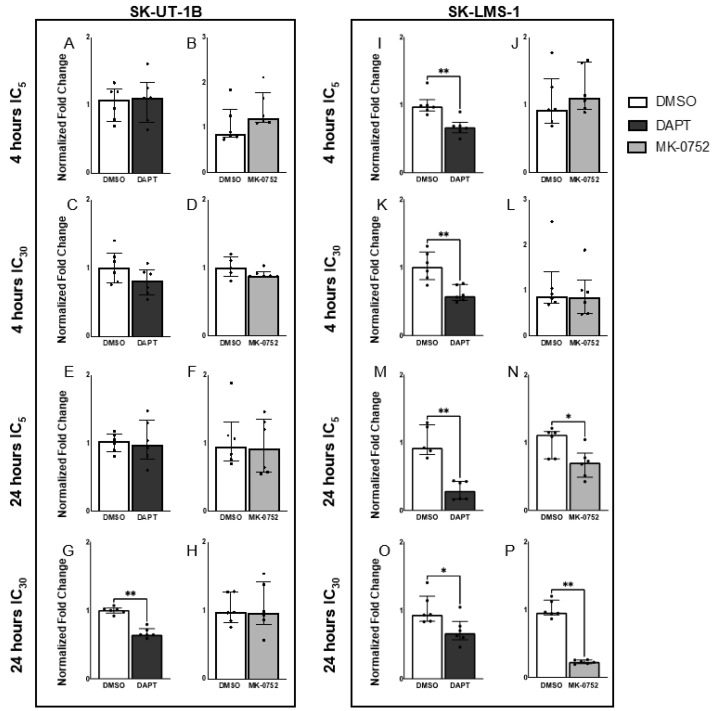
Expression of *HES1* in uLMS cells after treatment with DAPT and MK-0752. qRT-PCR determination of *HES1* expression after exposure to DAPT, MK-0752, or the DMSO control. The relative expression level of *HES1* was compared to *18s* rRNA. Expression of *HES1* in SK-UT-1B after 4 h of exposure to DAPT at IC_5_ of 15 µM (**A**), DAPT at IC_30_ of 50 µM (**C**), MK-0752 at IC_5_ of 50 µM (**B**) or MK-0752 at IC_30_ of 95 µM (**D**) was similar to the DMSO controls. Expression of *HES1* in SK-UT-1B after 24 h of exposure to DAPT at IC_5_ of 15 µM (**E**), MK-0752 at the IC_5_ of 50 µM (**F**) or MK-0752 at the IC_30_ of 95 µM (**H**) was similar to the DMSO controls. *HES1* expression was significantly decreased in SK-UT-1B after 24 h with DAPT at the highest concentration, IC_30_ of 50 µM ((**G**), FC: 1.55, *p* ≤ 0.01). Expression of *HES1* was significantly decreased in SK-LMS-1 after 4 h with DAPT at IC_5_ of 20 µM ((**I**), FC:1.46, *p* ≤ 0.01) and IC_30_ of 60 µM ((**K**), FC:1.74, *p* ≤ 0.01). Expression of *HES1* in SK-LMS-1 after 4 h of exposure to MK-0752 at the IC_5_ of 50 µM (**J**) or MK-0752 at the IC_30_ of 100 µM (**L**) was similar to the DMSO controls. Expression of *HES1* in SK-LMS-1 was significantly decreased after 24 h of exposure to DAPT at IC_5_ of 20 µM ((**M**), FC: 3.17, *p* ≤ 0.01) and DAPT at IC_30_ of 60 µM ((**O**), FC: 1.40, *p* ≤ 0.05) and 24 h of exposure to MK-0752 at IC_5_ of 50 µM ((**N**), FC: 1.60, *p* ≤ 0.05) and MK-0752 at IC_30_ of 100 µM ((**P**), FC:4.17, *p* ≤ 0.01). Data are presented as median + IQR with *n* = 6 for each treatment and * *p* ≤ 0.05, ** *p* ≤ 0.01.

## Data Availability

Not applicable.

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
