# Peer review of "Gamma Secretase Inhibitors as Potential Therapeutic Targets for Notch Signaling in Uterine Leiomyosarcoma"

_ijms, 2022, doi:10.3390/ijms23115980_

Round 1

Reviewer 1 Report

The current manuscript entitled “Gamma Secretase Inhibitors as Potential Therapeutic Targets for Notch Signaling in Uterine Leiomyosarcoma” demonstrates interesting issues about the implication of Notch in signaling pathways in uLMS. The current manuscript is well written and highlights the importance of the presented study. However, there are some minor weaknesses that need to be addressed before this research article becomes acceptable for publication. For example, in introduction, the authors could add further information about the main aspects of their research in order to give more emphasis on its significance. Additionally, the content of the study should be extensively reviewed by the authors for grammatical, syntax errors as well as typos.

Reviewer 2 Report

Many cytotoxic regimens have been tested, and the majority of studies have used doxorubicin, ifosfamide, gemcitabine, docetaxel, trabectedin, dacarbazine and pazopanib as single agents or in combination. The response rate by chemotherapy ranges approximatively from 10 to 50%, better especially for combination regimes, but the prognosis of this disease still remains poor. Therefore, the discovery of novel, more effective targeted treatments on the basis of molecular profiling together with the identification of prognostic molecular markers remains an unmet clinical need.

 Based on these premises, the paper addresses a very timely and important topic in this setting.

We would recommend some changes:

  • Introduction section: although the authors correctly included important papers in this setting, we believe they should add more details to the therapeutic background of uLMS and some recent studies should be cited within the introduction ( PMID: 32751892 ; PMID: 31417921 ; PMID: 32674439), only for a matter of consistency. We think it might be useful to introduce the topic of this interesting study.
  • Methods and Statistical Analysis: nothing to add.
  • Discussion section: Very interesting and timely discussion. Of note, the authors should expand the Discussion section, including a more personal perspective to reflect on. For example, they could answer the following questions – in order to facilitate the understanding of this complex topic for readers: what potential does this study hold? What are the knowledge gaps and how do researchers tackle them? How do you see this area unfolding in the next 5 years? We think it would be extremely interesting for the readers.

However, we think the authors should be acknowledged for their work. In fact, they correctly addressed an important topic, the methods sound good and their discussion is well balanced.

One additional little flaw: the authors could better explain the limitations of their work, in the last part of the Discussion.

We believe this article is suitable for publication in the journal although major revisions are needed. The main strengths of this paper are that it addresses an interesting and very timely question and provides a clear answer, with some limitations.

We suggest the addition of some references for a matter of consistency. Moreover, the authors should better clarify some points

Round 2

Reviewer 2 Report

Acceptance.